# OD³: Optimization-free Dataset Distillation for Object Detection

## Abstract

Training large neural networks on large-scale datasets requires substantial computational resources, particularly for dense prediction tasks such as object detection. Although dataset distillation (DD) has been proposed to alleviate these demands by synthesizing compact datasets from larger ones, most existing work focuses solely on image classification, leaving the more complex detection setting largely unexplored. In this paper, we introduce OD³, a novel optimization-free data distillation framework specifically designed for object detection. Our approach involves two stages: first, a candidate selection process in which object instances are iteratively placed in synthesized images based on their suitable locations, and second, a candidate screening process using a pre-trained observer model to remove low-confidence objects. We perform our data synthesis framework on MS COCO and PASCAL VOC, two popular detection datasets, with compression ratios ranging from 0.25% to 5%. Compared to the prior solely existing dataset distillation method on detection and conventional core set selection methods, OD³ delivers superior accuracy, establishes new state-of-the-art results, surpassing prior best method by more than 14% on COCO $mAP_{50}$ at a compression ratio of 1.0%. The code is in the supplementary material.

## 1. Introduction

Deep neural networks have achieved remarkable performance across a wide range of computer vision tasks (He et al., 2016; Ren, 2015; Dosovitskiy, 2020; Kirillov et al., 2023), but training these models generally requires substantial computational and data resources. Conventional strategies often involve collecting increasingly large datasets (Deng et al., 2009) and training ever larger networks (Dehghani et al., 2023) to capture data complexity. This paradigm is particularly evident in object detection (Shao et al., 2019), where the need for rich annotations, such as bounding boxes or even instance masks, can greatly increase dataset sizes and labeling overhead. As a result, there is a growing interest in techniques that enable

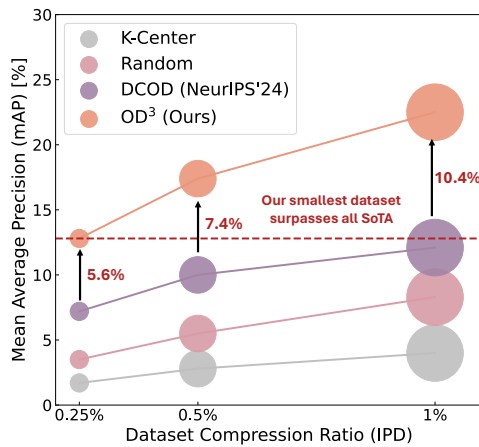

*Figure 1.* **Performance of OD³ and other methods on COCO dataset.** This illustration compares the mAP performance of our OD³ method with others on COCO with different compression ratios (IPDs), with an upper bound of 39.8% on the full dataset.

the creation of smaller, more manageable datasets capable of approximating the performance achieved by training on the original data. One promising direction in this area is dataset distillation (DD), which aims to synthesize *condensed* datasets that are significantly smaller yet still effective for training.

The majority of DD approaches have focused on image classification, where each image contains an object or a dominant label. This narrow scope overlooks the complexity and diversity of more demanding tasks, specifically object detection. In contrast to classification, object detection requires localizing and identifying multiple instances of potentially different classes in a single image. This jump in task complexity involves learning a mapping from image to label and predicting bounding boxes and class labels for multiple regions within the same image. Consequently, methods that successfully distill datasets for classification often struggle to adapt to the richer problem space of detection.

Another critical distinction lies in the type of supervision and evaluation metrics used in object detection tasks. While classification tasks use labels that can be applied at the image level, detection tasks rely on spatial annotations that align individual objects to bounding boxes, complete with class labels. This requirement introduces additional challenges when creating distilled datasets, as both the geometry

*Figure 2.* **Illustration of the OD³ framework.** In initial stage ①, each object in $\mathbf{x}_i \in \mathcal{T}$ is assigned a random location in the synthesized image $\hat{\mathbf{x}}_j \in \mathcal{S}$ for $j = 1, ..., \texttt{IPD}$, and its overlap with existing candidates is checked to decide placement. After IPD synthesized images are initially constructed, a pre-trained observer model produces predictions for screening. The observer takes the canvas from the previous cycle as input, analyzing the entire canvas, including the relationships between different objects and the specific objects present. These factors influence the output of ② *candidate screening*. The observer iteratively evaluates the current canvas to identify and remove objects that do not meet expectations, maintaining alignment with the post-evaluation process. For final reconstruction, the objects are inserted using their bounding boxes into $\hat{\mathbf{x}}_j \in \mathcal{S}$. Post-evaluation of $\mathcal{S}$ is carried out by fast distilling (Shen & Xing, 2022) knowledge from the observer model to a target network using PKD (Cao et al., 2022) loss on the respective feature pyramid networks.

(location) and identity (class) of objects must be preserved or effectively synthesized. Approaches that merely compress high-level category information may fail to capture the crucial spatial relationships and visual diversity that define detection tasks.

From a cost perspective of synthesis time, optimization-free approaches of DD prioritize efficiency and simplicity, bypassing computationally intensive iterative processes. One example is RDED (Sun et al., 2024), which generates synthetic classification datasets by directly extracting and combining realistic patches from original data. Using pre-trained models, it identifies informative crops, ensuring both diversity and structural integrity. This strategy shows the practicality and effectiveness of optimization-free methods.

In light of these complexities, we propose **O**ptimization-free **D**ataset **D**istillation for Object **D**etection (OD³), a novel framework explicitly tailored to address the unique challenges of synthesizing small, high-fidelity datasets for object detection. The framework leverages instance-level labels with scale-aware dynamic context extension (SA-DCE) to reconstruct diverse training images guided by an observer model, which is grounded in two core ideas: (1) an iterative *candidate selection* process that strategically places object instances in synthesized images, and (2) a *candidate screening* process powered by a pre-trained observer model,

which discards low-confidence objects. By removing the need for complex optimization procedures in constructing these synthetic images, OD³ provides a more streamlined and adaptable approach to DD for dense prediction tasks. The main contributions of this work are as follows:

- We propose a novel DD framework specifically designed for object detection, named OD³. It involves a two-stage process: *candidate selection*, where masked objects are localized and selected based on minimal overlap, and *candidate screening*, where a pre-trained observer filters unreliable candidates.

- OD³ bridges a crucial gap by extending the concept of dataset distillation beyond the relatively well-explored territory of image classification to the more challenging domain of object detection in a *training-free scheme*. Through a carefully designed process that handles both the spatial and semantic requirements of detection, our framework enables significant reductions in dataset size without sacrificing performance significantly.

- We evaluate our framework on MS COCO with compression ratios ranging 0.25% to 5% and on PASCAL VOC from 0.5% to 2.0%. The results demonstrate that our framework effectively reduces dataset size while maintaining model accuracy, providing an efficient solution for training object detectors.

## 2. Related Work

### 2.1. Dataset Distillation for Image Classification

**Bi-level optimization methods.** Bi-level optimization methods in dataset distillation tackle a nested problem: the inner loop trains a model on the distilled dataset, while the outer loop optimizes the dataset. Key approaches include FRePo (Zhou et al., 2022) to utilize neural feature regression with pooling, gradient matching (Kim et al., 2022) to align training behavior, feature matching (Wang et al., 2022) to preserve structural information, distribution matching (Zhao & Bilen, 2023) to align statistical properties, and trajectory matching (Cazenavette et al., 2022) to mimic the learning process. While these methods capture essential data characteristics, their high computational cost has spurred interest in optimization-free alternatives.

**Uni-level Optimization Methods.** SRe$^2$L (Yin et al., 2024) introduces a new framework by decoupling the bi-level optimization into two single-level learning processes, enabling efficient data condensation for datasets with varying sizes and image resolutions. Follow-up works include CDA (Yin & Shen, 2024), G-VBSM (Shao et al., 2024a), LPLD (Xiao & He, 2024), EDC (Shao et al., 2024b), LWTI (Qin et al., 2024), etc. Uni-level optimization methods streamline the dataset optimization process by eliminating nested optimization loops. This makes them well-suited for handling large-scale datasets while maintaining efficiency and scalability. Finally, an *optimization-free paradigm* was proposed in RDED (Sun et al., 2024), which provides an efficient approach that prioritizes both the realism and diversity.

### 2.2. Dataset Distillation for Object Detection

**Core-set Selection.** Coreset selection has emerged as one solution for reducing dataset size, primarily in image classification. It shows challenges in object detection, where multiple objects may appear in a single image. Recently, CSOD (Lee et al., 2024) introduces Coreset Selection for Object Detection, which selects image-wise and class-wise representative features for multiple objects of the same class using submodular optimization. Similarly, Training-Free Dataset Pruning (Anonymous, 2024) addresses dataset pruning for instance segmentation, tackling pixel-level annotations and class imbalances without training. However, these methods often achieve low compression ratios, typically above 20%. In contrast, our proposed distillation method compresses the original dataset to 0.5% or less.

**Dataset Disitllation.** Currently, efforts in dataset distillation for object detection remain limited. The first framework DCOD (Qi et al., 2024) was proposed for this purpose. DCOD employs a two-stage process: *Fetch* and *Forge*. During the *Fetch* stage, an object detection model is trained on the original dataset to extract essential features for localization and recognition tasks, similar to the squeezing process in SRe$^2$L (Yin et al., 2024). In the *Forge* stage, synthetic images are generated via model inversion, embedding required information into the images through uni-level optimization.

## 3. Method

### 3.1. Preliminaries

**Dataset Distillation for Object Detection.** The goal of `OD`$^3$ is to compress a large object detection dataset $\mathcal{T} = \{(\mathbf{x}_i, \{< \boldsymbol{b}_{i1}, \boldsymbol{c}_{i1}, \dots >\})\}$ $(i = 1, \dots, |\mathcal{T}|)$ into a much smaller synthesized dataset $\mathcal{S} = \{(\hat{\mathbf{x}}_j, \{< \hat{\boldsymbol{b}}_{j1}, \hat{\boldsymbol{c}}_{j1}, \dots >\})\}$ $(j = 1, \dots, |\texttt{IPD}|)$ that maintains the significant characteristics of $\mathcal{T}$ in terms of overall performance, where $\boldsymbol{b} = \{\mathbf{x}_c, \mathbf{y}_c, \boldsymbol{w}, \boldsymbol{h}\}$ represents the center of the bounding box and the width and height of the image. Here, $|\mathcal{S}| \ll |\mathcal{T}|$ and $\texttt{IPD}$ is the notion of images per dataset which reflects the compression ratio[1]. The performance of a model with weights $\theta_\mathcal{S}$ trained on $\mathcal{S}$ should be similar to that of a model with weights $\theta_\mathcal{T}$ trained on $\mathcal{T}$, within a small margin $\boldsymbol{\epsilon}_{DD}$. This can be expressed as:

$$sup\{|\mathcal{L}_{\theta_\mathcal{T}} - \mathcal{L}_{\theta_\mathcal{S}}|\}_{(\mathbf{x}_v, \mathbf{y}_v) \in \mathcal{T}'} \leq \boldsymbol{\epsilon}_{DD} \qquad (1)$$

with $\mathcal{L}$ representing the loss function, and $(\mathbf{x}_v, \mathbf{y}_v) \in \mathcal{T}'$ is some test or val set associated with $\mathcal{T}$.

**Definition 3.1** (Optimization-free dataset distillation for object detection). *Our objective is to collect as much effective information as possible on a "blank canvas", interpreted as an initially empty image. The information is considered "effective" if it contains sufficient high-quality (high-confidence, well-sized) objects.*

**Information Density.** To quantify how thoroughly a canvas is occupied by valuable objects, we define an *Information Density* function $\Phi(\mathbf{x})$:

$$\Phi(\mathbf{x}) = \frac{g(f_\theta(\mathbf{x}))}{a(\mathbf{x})}, \qquad (2)$$

where $\mathbf{x}$ is the current canvas (image) under consideration. $f_\theta(\cdot)$ is a well-trained object detector parameterized by $\theta$. $g(\cdot)$ is a function that aggregates detection confidence scores across all detected objects. $a(\mathbf{x})$ denotes the combined area of all detected objects on $\mathbf{x}$.

Concretely, we instantiate $g(\cdot)$ and $a(\cdot)$ as follows:

$$\Phi(\mathbf{x}) = \frac{\sum_{r=0}^{K} a(\boldsymbol{o}_r) q(\boldsymbol{o}_r)}{\sum_{r=0}^{K} a(\boldsymbol{o}_r)}, \qquad (3)$$

where $K$ is the total number of objects placed on the canvas $\mathbf{x}$, $\boldsymbol{o}_r$ is the $r$-th object, $a(\boldsymbol{o}_r)$ represents the area of object $\boldsymbol{o}_r$, $q(\boldsymbol{o}_r)$ is the detector-derived confidence score for $\boldsymbol{o}_r$[2].

---

[1]We define `IPD` (images per dataset) instead of conventional `IPC` (images per class) used in classification task as in object detection each image can contain multi-object with different classes.

[2]In our paper, $i$, $j$ and $r$ are the image index of original dataset, index of distilled image, and object index, respectively.

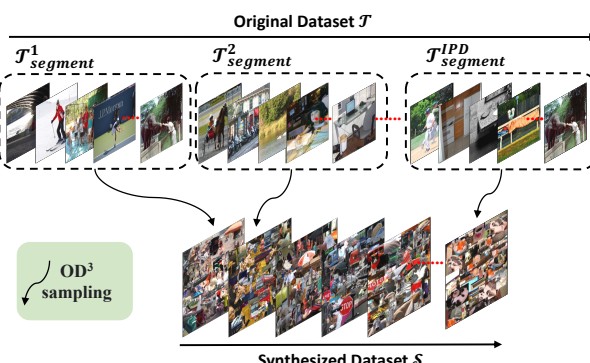

**Original Dataset $\mathcal{T}$**

$\mathcal{T}^1_{segment}$  $\mathcal{T}^2_{segment}$  $\mathcal{T}^{IPD}_{segment}$

OD³ sampling

**Synthesized Dataset $\mathcal{S}$**

*Figure 3.* **Illustration of our sampling controller.** It ensures that the same object is not placed in different distilled images. The original dataset is divided into IPD segments. Each segment is distilled into a single image, resulting in a compact dataset $\mathcal{S}$.

Thus, $\Phi(\mathbf{x})$ measures how confidently and extensively the canvas is occupied by objects.

**Information Diversity.** In addition to confidence and size, we also encourage diversity of objects on the canvas. We define a simple *Information Diversity* $\mathcal{N}(\mathbf{x})$ by:

$$\mathcal{N}(\mathbf{x}) = N. \tag{4}$$

where $N$ is the number of distinct objects on the canvas $\mathbf{x}$. Even when a few objects exhibit high confidence, having more *distinct* objects can yield richer training signals, making the distilled data more robust.

### 3.2. OD³ Framework

**Overview.** Unlike prior dataset distillation methods, our approach begins with a *blank canvas* as the starting point for generating each new synthetic data sample. As shown in Fig. 2, the data distillation process first proceeds with a candidate selection stage (orange box, bottom-left), where object instances are extracted from an existing large-scale dataset $\mathcal{T}$. For each image $\mathbf{x}_i \in \mathcal{T}$, bounding boxes $\{\boldsymbol{b}_{i1}, \boldsymbol{b}_{i2}, \ldots, \boldsymbol{b}_{iK}\}$ ($K$ is the number of bounding boxes) capture potentially useful object patches. These patches are fed into a localization operation, a random yet controlled placement mechanism that carries out $M$ attempts of inserting each candidate onto a reconstructed canvas without exceeding the overlap threshold. Fig. 3 shows the sampling strategy that ensures that $|\mathcal{S}| = $ IPD and that objects in $\mathcal{S}$ are all unique. This yields a large *pool of candidate patches* $(\boldsymbol{b}_i, \boldsymbol{l})$ on the canvas, where $\boldsymbol{l}$ is the bounding box label or class. Our illustration also highlights how some patches that fail overlap constraints are discarded.

Once a preliminary reconstructed image is assembled, the process proceeds to the candidate screening / filtering stage. Here, an observer model (a pre-trained detector) performs inference on the partially reconstructed canvas. Its predictions are matched with ground-truth boxes that originated

---

**Algorithm 1** **O**ptimization-free **D**ataset **D**istillation for **O**bject **D**etection (**OD³**)

**Input:** Original dataset $\mathcal{T}$; Synthetic dataset $\mathcal{S}$; Observer model $\theta_{\text{obs}}$; Overlap threshold $\tau$; Screening threshold $\eta$; Images per dataset IPD; Canvas $\mathcal{C}$ (initially $\varnothing$ and updated constantly with $\hat{\mathbf{x}}$); Extension $\boldsymbol{\ell}$ in Eq. 7; Random placement candidate positions $\langle \mathbf{m}_t, \mathbf{n}_t \rangle$ for $t = 1, \ldots, \mathbf{M}$.

**for** $\hat{\mathbf{x}}_j \in \mathcal{S}$ *where* $|\mathcal{S}| = $ IPD **do**
  **while** $\mathcal{C}$ *is not full* **do**
    ▷ Candidate Selection & Placement
    **for** $(\mathbf{x}_i, \mathbf{y}_i) \in \mathcal{T}$ **do**
      **for** $< \boldsymbol{b}_{ir}, \boldsymbol{c}_{ir} > \in \mathbf{y}_i$ **do**
        $\boldsymbol{b}'_{ir} \leftarrow \boldsymbol{b}_{ir} + \boldsymbol{\ell}_{ir}$ ▷ $\boldsymbol{\ell}_{ir} \leftarrow F(\bar{r})$
        **while** $IoU(\boldsymbol{b}'_{ir}, \langle \mathbf{m}_t, \mathbf{n}_t \rangle, \mathcal{C}) < \tau$
        **and** *attempts* $< \mathbf{M}$ **do**
          Place $< \boldsymbol{b}'_{ir}, \boldsymbol{c}_{ir} > \rightarrow \mathcal{C}$; Exit

    ▷ Candidate Screening
    Filter objects from $\mathcal{C}$ for $\hat{\mathbf{x}}$
    $\hat{\mathbf{y}}_{\text{obs}} = \theta_{\text{obs}}(\mathcal{C})$
    **for** $< \boldsymbol{b}_k, \boldsymbol{c}_k > \in \mathcal{C}$ **do**
      **if** $Conf(\hat{\mathbf{y}}_{obs}, \boldsymbol{b}_k) < \eta$ **then**
        Remove $< \boldsymbol{b}_k, \boldsymbol{c}_k >$ from $\mathcal{C}$
    $\hat{\mathbf{x}}_j \leftarrow \mathcal{C}$
  Append $\hat{\mathbf{x}}_j$ to $\mathcal{S}$
**Output:** Synthesized dataset $\mathcal{S}$

---

from the bounding boxes inserted into the image. Objects that fail to meet confidence or consistency criteria are removed, refining the canvas into a high-quality, diversified arrangement of objects. As a result, the final reconstructed image $\hat{\mathbf{x}}_j \in \mathcal{S}$ now contains only those patches that pass the screening process. Also, the bounding box and class annotations associated with these patches are transformed into soft labels, enabling more nuanced supervision in subsequent stages. The synthesis process is presented in Algorithm 1, and we elaborate on the details of each step below.

**Iterative Copy-paste and Removal Process.** First, each object candidate is added to the partially formed "blank canvas" via `random copy-paste`. Multiple objects may be overlaid, so that visual variety is preserved. Next, the observer model runs on this synthesized image and assesses the confidence of each placed object. Low-confidence objects that are not matched to the ground truth objects are removed, refining the canvas into a more coherent scene. This cycle of *add-and-remove* iterates multiple times, driving the canvas toward a final state containing only high-confidence, mid-overlapping objects. Fig. 2 green boxes in screening stage indicate an inserted object is deemed infeasible, applying removal process to maintain quality and coherence.

**Soft Label Generation.** In image classification, the concept

of soft labels serves as a foundational element in constructing condensed datasets, yielding substantial performance gains, as noted by (Yin et al., 2024). Logit-based soft labels, in particular, play a critical role in improving the generalization capability of validation models trained on condensed datasets through KD framework (Hinton, 2015). However, when applied to object detection tasks, logit-based soft labels fail to deliver competitive accuracy. This raises the necessity of developing a specialized soft label design tailored explicitly for dataset distillation in object detection. The most typical kind of soft label used in object detection is the output of the feature pyramid network (FPN). This output $\mathbf{y}^{\text{feat}}$ can be defined as $\mathbb{R}^{C \times H \times W}$, where $C$, $H$ and $W$ represent the number of channels, the height of the canvas and the width of canvas, respectively. Once the (feature-based) soft label $\{\mathbf{y}_i^{\text{feat}}\}$ has been obtained, it is employed during the post-evaluation phase and supervised using the following loss function (Shu et al., 2021):

$$\mathcal{L}_{\text{mse}} = \mathbb{E}_{(\mathbf{x}_i, \mathbf{y}_i^{\text{feat}})} \|\mathbf{y}_i^{\text{feat}} - f^{\text{fpn}}(f^{\text{backbone}}(\mathbf{x}_i))\|_2^2, \quad (5)$$

where $\{(\mathbf{x}_i, \mathbf{y}_i^{\text{feat}})\}$, $f^{\text{fpn}}$ and $f^{\text{backbone}}$ refer to the condensed dataset, the FPN in the model and the backbone of the model, respectively. However, we observe that this form of soft label is hard to provide sufficient information for detection.

Thus, we consider a channel-wise soft label for enhancing the performance of the evaluated detector. We leverage the simple pearson knowledge distillation (PKD) (Cao et al., 2022) as a basis for designing the soft label generation mechanism on object detection. Given this, we can give the form of soft label as $\left\{ \frac{f^{\text{fpn}}(f^{\text{backbone}}(\mathbf{x}_i)) - \text{mean}(f^{\text{fpn}}(f^{\text{backbone}}(\mathbf{x}_i)))}{\text{std}(f^{\text{fpn}}(f^{\text{backbone}}(\mathbf{x}_i))) + \epsilon} \right\}$, where $\text{mean}(\cdot)$, $\text{std}(\cdot)$ and $\epsilon$ denote the mean operator in the height and width dimensions, the standard deviation operator in the height and width dimensions, and very small amounts, respectively. Finally, the condensed dataset and its associated soft labels are used to train a new detector initialized randomly to test how well this small synthetic dataset supports the downstream detection task. As shown in the post-evaluation stage, the condensed dataset supervises the target detector training, and the PKD used in post-evaluation can be formulated as:

$$\mathcal{L}_{\text{mse}} = \mathbb{E}_{(\mathbf{x}_i, \mathbf{y}_i^{\text{feat}})} \left\| \mathbf{y}_i^{\text{feat}} - \frac{f^{\text{fpn}}(f^{\text{backbone}}(\mathbf{x}_i)) - \text{mean}(f^{\text{fpn}}(f^{\text{backbone}}(\mathbf{x}_i)))}{\text{std}(f^{\text{fpn}}(f^{\text{backbone}}(\mathbf{x}_i))) + \epsilon} \right\|_2^2, \quad (6)$$

**Scale-aware Dynamic Context Extension.** We also propose a simple *scale-aware dynamic context extension* (SA-DCE) for varying sizes of objects in detection-based dataset distillation as a crucial enhancement that directly addresses the challenges posed by small objects with limited contextual information. Unlike the optimization-based method (Qi et al., 2024), which struggles to preserve or amplify contextual cues due to their reliance on fixed gradients and

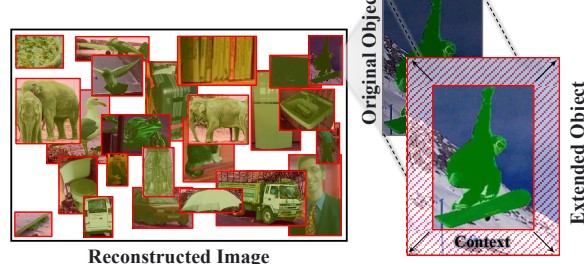

**Figure 4. Extended bounding box for more object context.** The figure shows a reconstructed image along with an extended object using the SA-DCE function to better capture the object's context.

pixel-specific updates, context extension involves intentionally expanding the bounding region around objects. It can be formulated as a function of the object's size:

$$\boldsymbol{\ell}_{\text{extension}} = F(\boldsymbol{o}_{i_r}, \overline{\boldsymbol{r}}) = \left(1 - \frac{a(\boldsymbol{o}_{i_r}) - \overline{a}_{\min}}{\overline{a}_{\max} - \overline{a}_{\min}}\right) \times \overline{\boldsymbol{r}}, \quad (7)$$

where $F$ is the SA-DCE function, $\overline{\boldsymbol{r}} \in \mathbb{R}$ is a scalar representing a small pre-determined number of pixels, $a(\boldsymbol{o}_{i_r})$ is the area of $r$-th object in $i$-th image, and $\overline{a}_{\max}$ and $\overline{a}_{\min}$ represent the maximum and minimum areas of objects in $\mathcal{T}$. An example of SA-DCE can be seen in Fig. 4.

This subtle yet impactful modification adds peripheral context that is often missing, especially in small object representations, providing the model with additional spatial cues that help in accurate detection. By extending the context, our model can better differentiate objects from the background, leading to improved performance, particularly in complex scenes. Optimization-based methods inherently lack the flexibility to incorporate such targeted context adjustments, as they are confined to the synthetic data representation derived from iterative pixel tuning.

**Objective.** We combine the two metrics in Eq. 3 and Eq. 4 into a single objective for data distillation:

$$\mathcal{S}_{\hat{\mathbf{x}}} = \underset{\mathbf{x}_T}{\arg\max} \ \ \Phi(\mathbf{x}_T) + \mathcal{N}(\mathbf{x}_T), \quad (8)$$

where $\mathbf{x}_T$ denotes the final condensed canvas (i.e., synthesized image) after $T$ synthesis iterations. In practice, we do not explicitly find their optimal values separately, as they are mutually restrictive and entangled. Once the size of the canvas is predefined, we can simply perform an ablation study on *the overlap of objects on canvas* for the optimal value that maximizes $\mathcal{S}_{\hat{\mathbf{x}}}$, as detailed in the following section.

During the iterative data-synthesis process, we update $\mathbf{x}_i$ for $i = 0, 1, \ldots, T - 1$ using:

$$\mathbf{x}_{i+1} = f_{\text{remove}}(f_{\text{add}}(\mathbf{x}_i)), \quad i \in 0, 1, 2, \ldots, T - 1. \quad (9)$$

Here, $f_{\text{add}}(\cdot)$ adds new candidate objects to the current canvas, $f_{\text{remove}}(\cdot)$ filters out low-confidence or redundant objects, thereby refining the composition of $\mathbf{x}_i$.

**Iterative Synthesis Methods.** We consider two iterative processes for building the final canvas $\mathbf{x}_T$.

**1. First Form: Add-Only.** The process of this startegy is:

$$\mathbf{x}_{i+1} = f_{\text{add}}(\mathbf{x}_i), \quad i = 0, 1, \ldots, T-1 \qquad (10)$$

In this scenario, newly added objects remain on the canvas even if their confidence is low and if they overlap other objects smaller than $\tau$. The final objective value is

$$G_1 = \Phi(\mathbf{x}_T^a) + \mathcal{N}(\mathbf{x}_T^a) \qquad (11)$$

**2. Second Form: Add-Then-Remove.** The *Add-Then-Remove* is a loop to construct then refine distilled images:

$$\mathbf{x}_{i+1} = f_{\text{remove}}(f_{\text{add}}(\mathbf{x}_i)), \quad i = 0, 1, \ldots, T-1 \quad (12)$$

Here, each iteration first adds new objects, then filters out objects whose confidence $q(o_j)$ is below a threshold $\eta$, or that fail other criteria (e.g., excessive overlap). The final objective value is

$$G_2 = \Phi(\mathbf{x}_T^{ar}) + \mathcal{N}(\mathbf{x}_T^{ar}) \qquad (13)$$

The following theorem states that **incorporating the remove step** will positively increases the objective, under enough iterations and a well-chosen confidence threshold.

**Theorem 3.2.** *(the proof in Appendix C) Under the above definitions, we have*

$$G_2 \geq G_1. \qquad (14)$$

**Intuition.** Because adding a removal step $f_{\text{remove}}(\cdot)$ after every object-addition $f_{\text{add}}(\cdot)$ enables a more refined composition of the canvas, the *second form* is guaranteed to achieve at least as high an objective value as the simpler first form (which lacks a removal step). That is, the second iterative scheme (*add-then-remove*) achieves an objective value at least as large as the add-only approach, under typical assumptions on how objects are added or removed.

*Sketch Proof.* Setup: $f_{\text{remove}}$ is an operator that detects objects in the canvas $\mathbf{x}_i$ and removes those with confidence below a threshold $\eta$. Concretely:

*Step-1: Detection step.* Compute $f_\theta(\mathbf{x}_i)$, i.e., run a pre-trained detector on the current canvas $\mathbf{x}_i$.

*Step-2: Scoring each object.* For each object $o_{i_r}$ in $\mathbf{x}_i$ (where $r = 1, \ldots, K$, $K$ is the number of objects), obtain a confidence score $q(o_{i_r})$.

*Step-3: Threshold partition (no overlaps assumed).* Divide the objects into two groups $\mathcal{O}_1$ and $\mathcal{O}_2$, one with a confidence level greater than the threshold $\eta$, and the other with a confidence level less than or equal to the threshold $\eta$:

$$\mathcal{O}_1 := \{o_{i_0}, o_{i_1}, \ldots, o_{i_M}\}, \text{ where } q(o_{i_0}) \leq \cdots \leq q(o_{i_M}) < \eta$$

$$\mathcal{O}_2 := \{o_{i_{M+1}}, o_{i_{M+2}}, \ldots, o_{i_K}\},$$

$$\text{where } \eta \leq q(o_{i_{M+1}}) \leq q(o_{i_{M+2}}) \leq \cdots \leq q(o_{i_K}) \qquad (15)$$

| IPD | Method ↓ | mAP (%) | mAP$_{50}$ (%) | mAP$_{75}$ (%) |
|---|---|---|---|---|
| 0.25% | Random | 3.50 | 9.70 | 1.60 |
| | Uniform | 3.60 | 9.80 | 1.60 |
| | K-Center | 1.70 | 6.30 | 0.40 |
| | Herding | 1.70 | 5.80 | 0.50 |
| | DCOD (Qi et al., 2024) | 7.20 | 17.20 | 4.80 |
| | **OD$^3$ (Ours)** | **12.80**$_{(+5.6)}$ ▲ | **24.70**$_{(+7.5)}$ ▲ | **11.90**$_{(+7.1)}$ ▲ |
| 0.5% | Random | 5.50 | 14.20 | 2.90 |
| | Uniform | 5.60 | 14.30 | 2.90 |
| | K-Center | 2.80 | 8.90 | 0.70 |
| | Herding | 2.60 | 8.80 | 0.80 |
| | DCOD (Qi et al., 2024) | 10.00 | 21.50 | 9.00 |
| | **OD$^3$ (Ours)** | **17.40**$_{(+7.4)}$ ▲ | **32.10**$_{(+10.6)}$ ▲ | **17.00**$_{(+8.0)}$ ▲ |
| 1.0% | Random | 8.30 | 19.70 | 5.30 |
| | Uniform | 8.40 | 19.70 | 5.40 |
| | K-Center | 4.00 | 12.90 | 1.20 |
| | Herding | 4.10 | 12.50 | 1.30 |
| | DCOD (Qi et al., 2024) | 12.10 | 24.70 | 10.40 |
| | **OD$^3$ (Ours)** | **22.50**$_{(+10.4)}$ ▲ | **39.60**$_{(+14.9)}$ ▲ | **22.90**$_{(+12.5)}$ ▲ |

Table 1. **Performance Comparison on MS COCO.** The compression ratios range from 0.25% to 1.0%. The observer model and the student model are Faster R-CNN 101 and Faster R-CNN 50.

*Step-4: Removing low-confidence objects.* All objects whose confidence $< \eta$ are discarded. Thus the *information density* on the canvas changes as follows:

$$\frac{\sum_{r=0}^K a(o_{i_r}) q(o_{i_r})}{\sum_{r=0}^K a(o_{i_r})} \longrightarrow \frac{\sum_{r=M+1}^K a(o_{i_r}) q(o_{i_r})}{\sum_{r=M+1}^K a(o_{i_r})}.$$

**Comparison of Densities.** To see why the new density (after removal) is generally higher or equal, we can interpret $\frac{o_{i_k}}{\sum_{r=0}^K a(o_{i_r})}$ as a probability weight, letting $o_{i_k}$ denote the area $\times$ score contribution of object $k$. Removing those objects whose confidence is below $\eta$ essentially removes low-quality (score or area) contributions from the numerator, thereby increasing the average or expected confidence. If $K$ is sufficiently large, we can consider: $\mathbb{E}_r[q(o_{i_r})]$ and $\mathbb{E}_{r \geq M+1}[q(o_{i_r})]$, a standard probabilistic argument shows that the expected confidence of the surviving set $\{o_{i_{M+1}}, \ldots, o_{i_K}\}$ is at least as high as that of the entire original set. Formally,

$$\mathbb{E}_{r \geq M+1}[q(o_{i_r})] \geq \mathbb{E}_r[q(o_{i_r})], \qquad (16)$$

which implies $\Phi(\mathbf{x}_T) \geq \Phi(\mathbf{x}_{T-1}) \geq \cdots \geq \Phi(\mathbf{x}_0)$ in the *add-then-remove* scheme.

By similar reasoning (via a probabilistic bound on whether the leftover portion remains undetected), one can show that the presence of overlaps does not harm the objective in the *add-then-remove* scheme. Hence, $G_2 \geq G_1$ even when overlaps are considered. More details are in our Appendix.

## 4. Experiments

**Experimental Setup.** We evaluate OD$^3$ with compression ratios ranging from 0.25% to 5% for MS COCO (Lin et al., 2014) and from 0.5% to 2% for PASCAL VOC (Everingham et al., 2010). We set the overlap threshold $\tau$ to 0.6 in the candidate selection stage, the confidence threshold $\eta$ to

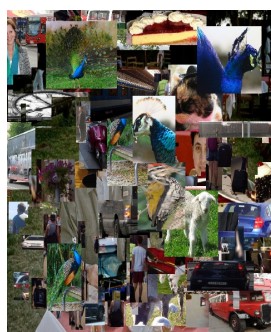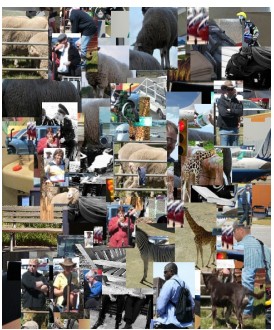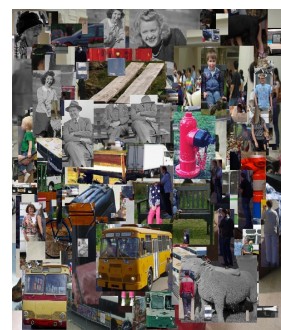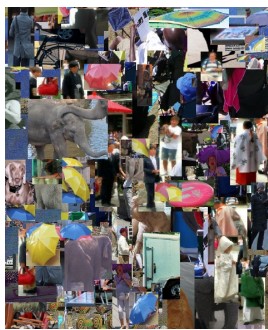

*Figure 5.* **Qualitative results of the synthesis process of OD³ on MS COCO.** Initial backgrounds of the canvas are randomly selected from the training set, and objects are inserted using their bounding-box level labels. Those images are generated at $0.5\%$ IPD.

| IPD | Method ↓ | mAP (%) | mAP$_{50}$ (%) |
|---|---|---|---|
| 0.5% | OD³ | 38.51 | 38.50 |
| 1.0% | OD³ | 51.05 | 51.10 |
| 2.0% | OD³ | 58.68 | 58.70 |

*Table 2.* **Performance on Pascal VOC.** The compression ratios range from $0.5\%$ to $2.0\%$. The observer model and the student model for our method are both Faster R-CNN 50.

| IPD | Label | mAP | mAP$_{50}$ | mAP$_{75}$ | mAP$_s$ | mAP$_m$ | mAP$_l$ |
|---|---|---|---|---|---|---|---|
| 0.25% | Mask | 10.90 | 20.80 | 10.30 | 3.50 | 15.10 | 15.90 |
| | Bbox | 12.40 | 23.90 | 11.60 | 4.90 | 16.10 | 18.10 |
| | **Ex-Bbox** | 12.80$_{(+0.4)}$▲ | 24.70$_{(+0.8)}$▲ | 11.90$_{(+0.3)}$▲ | 5.60$_{(+0.7)}$▲ | 16.80$_{(+0.7)}$▲ | 17.70$_{(-0.4)}$▼ |
| 0.5% | Mask | 15.10 | 28.00 | 14.80 | 5.60 | 20.30 | 22.30 |
| | Bbox | 16.60 | 30.30 | 16.30 | 6.80 | 21.70 | 23.30 |
| | **Ex-Bbox** | 17.40$_{(+0.8)}$▲ | 32.10$_{(+1.8)}$▲ | 17.00$_{(+0.7)}$▲ | 8.40$_{(+1.6)}$▲ | 23.00$_{(+1.3)}$▲ | 22.80$_{(-0.5)}$▼ |
| 1.0% | Mask | 21.20 | 37.40 | 21.30 | 8.90 | 26.40 | 29.90 |
| | Bbox | 22.00 | 38.70 | 22.30 | 9.70 | 27.40 | 30.10 |
| | **Ex-Bbox** | 22.50$_{(+0.5)}$▲ | 39.60$_{(+0.9)}$▲ | 22.90$_{(+0.6)}$▲ | 10.60$_{(+0.9)}$▲ | 28.00$_{(+0.6)}$▲ | 29.80$_{(-0.3)}$▼ |
| 5.0% | Mask | 30.00 | 49.50 | 31.70 | 15.10 | 34.80 | 39.10 |
| | Bbox | 29.90 | 49.40 | 31.50 | 15.00 | 34.70 | 38.50 |
| | **Ex-Bbox** | 30.10$_{(+0.2)}$▲ | 49.70$_{(+0.3)}$▲ | 31.80$_{(+0.3)}$▲ | 16.20$_{(+1.2)}$▲ | 34.90$_{(+0.2)}$▲ | 38.40$_{(-0.1)}$▼ |

*Table 3.* **Ablation Study on Label Type.** The impact of using mask labels, bounding box (BBox) labels, or Ex-BBox in the data synthesis process across various compression ratios. *Ex-Bbox* represents the BBox with the extended context using SA-DCE.

| IPD | Observer | Target | mAP (%) | mAP$_{50}$ (%) | mAP$_{75}$ (%) |
|---|---|---|---|---|---|
| 0.25% | RetinaNet | RetinaNet | 13.90 | 25.30 | 13.50 |
| | Faster R-CNN | RetinaNet | 14.50 | 25.70 | 14.30 |
| | Faster R-CNN | Faster R-CNN | 12.80 | 24.70 | 11.90 |
| 0.5% | RetinaNet | RetinaNet | 18.40 | 32.50 | 18.20 |
| | Faster R-CNN | RetinaNet | 17.40 | 30.20 | 17.60 |
| | Faster R-CNN | Faster R-CNN | 17.40 | 32.10 | 17.00 |
| 1.0% | RetinaNet | RetinaNet | 22.20 | 37.90 | 22.60 |
| | Faster R-CNN | RetinaNet | 22.20 | 37.40 | 23.00 |
| | Faster R-CNN | Faster R-CNN | 22.50 | 39.60 | 22.90 |
| 5.0% | RetinaNet | RetinaNet | 28.30 | 46.30 | 29.70 |
| | Faster R-CNN | RetinaNet | 30.00 | 48.60 | 31.10 |
| | Faster R-CNN | Faster R-CNN | 30.10 | 49.70 | 31.80 |

*Table 4.* **Cross-Architecture Performance Comparison for OD³ on MS COCO.** The results are evaluated across $0.25\%$ to $5.0\%$. Observer models use ResNet101 and target models use ResNet50.

0.2 in the candidate screening stage, and $\mathbf{M}$ to 40. The foreground objects are inserted into the reconstructed images using their ground truth bounding boxes with extended context using SA-DCE. The backgrounds of the reconstructed images are randomly sampled from the respective datasets.

| IPD | Candidate Selection | Candidate Screening | mAP | mAP$_{50}$ | mAP$_{75}$ | mAP$_s$ | mAP$_m$ | mAP$_l$ |
|---|---|---|---|---|---|---|---|---|
| 0.25% | ✗ | ✗ | 0.90 | 2.40 | 0.40 | 0.00 | 1.10 | 1.20 |
| | ✓ | ✗ | 9.70 | 19.10 | 8.90 | 3.70 | 13.10 | 13.60 |
| | ✓ | ✓ | **12.80** | **24.70** | **11.90** | **5.60** | **16.80** | **17.70** |
| 0.5% | ✗ | ✗ | 2.00 | 4.00 | 1.90 | 0.10 | 2.60 | 3.10 |
| | ✓ | ✗ | 14.50 | 27.30 | 13.80 | 5.90 | 19.30 | 20.30 |
| | ✓ | ✓ | **17.40** | **32.10** | **17.00** | **8.40** | **23.00** | **22.80** |
| 1.0% | ✗ | ✗ | 7.50 | 14.10 | 7.30 | 0.9 | 8.90 | 13.30 |
| | ✓ | ✗ | 19.00 | 33.90 | 19.10 | 8.70 | 24.40 | 25.70 |
| | ✓ | ✓ | **22.50** | **39.60** | **22.90** | **10.60** | **28.00** | **29.80** |
| 5.0% | ✗ | ✗ | 8.60 | 17.80 | 7.30 | 1.10 | 10.40 | 16.40 |
| | ✓ | ✗ | 28.10 | 46.70 | 29.60 | 14.00 | 34.00 | 37.00 |
| | ✓ | ✓ | **30.10** | **49.70** | **31.80** | **16.20** | **34.90** | **38.40** |

*Table 5.* **Ablation Study on Method Components.** We highlight the impact of candidate selection and screening on MS COCO performance across varying compression rates.

Synthesis experiments are conducted on a single 4090 GPU.

The canvas sizes used are $484 \times 578$ for MS COCO and $375 \times 500$ for PASCAL VOC, which are the average width and height of the respective full training sets. For the post-evaluation stage, we use VOC2007 and VOC2012 train/val splits combined for synthesis and VOC2007 test set for evaluation. We use standard COCO metrics ($mAP$, $mAP_{50}$, and $mAP_{75}$) along with size metrics ($mAP_s$, $mAP_m$, and $mAP_l$) for the COCO dataset. We use Pascal VOC style $mAP$ and $mAP_{50}$ with the area method that uses all points in the precision-recall curve instead of only 11, which provides a more precise evaluation (Everingham et al., 2010). Faster R-CNN-50 models are trained for 96 epochs and the RetinaNet-50 models for 256 epochs. All post-evaluation experiments are conducted on $2\times$ 4090 GPUs. Our implementation is based on the mmdetection (Chen et al., 2019) and mmrazor (Contributors, 2021) frameworks.

**Image Generation Time and Efficiency.** Our synthesis process does not require training, making data generation highly efficient compared to optimization-based approaches like DCOD (which did not report generation time). Our primary time overhead comes from screening by the observer. Specifically, generating the condensed dataset takes approximately 4.7 hours on MS COCO and 0.74 hours on PASCAL VOC using a single 4090 GPU.

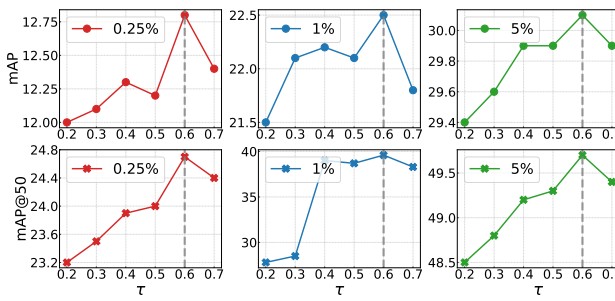

Figure 6. **Ablation study of overlap threshold $\tau$.** mAP and $mAP_{50}$ are evaluated at different thresholds used in candidate selection with compression ratios ranging from 0.25% to 5%.

## 4.1. Experimental Results

Table 1 presents the comparative results of our method on MS COCO (Lin et al., 2014) with core-set selection methods and with DCOD (Qi et al., 2024). The core-set selection methods include: random initialization (Rebuffi et al., 2017), Uniform (Lee et al., 2024), K-center (Sener & Savarese, 2017), and Herding (Castro et al., 2018; Chen et al., 2012). Our method, OD³, outperforms all other methods across various compression ratios (IPD) ranging from 0.25% to 1.0%. Notably, at 1.0%, we achieve a substantial 14.9% improvement in $mAP_{50}$ over DCOD. Furthermore, our method consistently outperforms other core-set selection methods, with $mAP_{50}$ improvements of up to 27.1% at 1.0% IPD. Our method also achieves the highest performance in $mAP$, $mAP_{50}$ and $mAP_{75}$ at each compression ratio. Since the authors of DCOD did not report its performance on the size metrics of MS COCO, we are unable to compare the methods in that regard. Nonetheless, these results underline the effectiveness of OD³ in achieving superior performance across a range of compression ratios. We also report results on PASCAL VOC in Table 2 with different IPDs, where the method achieves 58.68% $mAP$ at 2.0% compression.

## 4.2. Ablation Studies

**Label type.** The type of label used when inserting the candidate objects into the synthesized image is studied in Table 3. We consider three types of labels: mask-level label, BBox-level label, and *Ex-Bbox*, which refers to a BBox with extended context using SA-DCE (refer to Sec.3.2). Using BBox labels outperforms the mask labels across all IPDs except for 5.0%, where their performance converges to a similar level. This is because Bbox labels preserves local context and surrounding environment of individual objects, providing models with additional cues for recognizing objects. Using *Ex-Bbox* further improves performance across all IPDs, where an improvement of 1.8% in $mAP_{50}$ can be seen at 0.5% compression. When specifically analyzing the size metrics, the extended context benefits small objects the most on the account of large objects, which bridges the

substantial gap between their detection performance.

**Overlap threshold.** Fig. 6 shows how different values of overlap thresholds $\tau$ in candidate selection affect the performance of our method across various compression ratios. It can be seen that 0.6 consistently optimizes $mAP_{50}$ and $mAP$. This value can be thought of as an optimal trade-off between $\Phi(\mathbf{x})$ and $\mathcal{N}(\mathbf{x})$.

**Cross-architecture generalization.** To assess the generalization capability of our condensed datasets, we conduct experiments with Faster R-CNN (Ren, 2015), a two-stage detector and RetinaNet (Lin, 2017), a one-stage detector. Table 4 shows that the distilled datasets can generalize in heterogeneous settings, where the observer model is a two-stage detector, and the target model is a one-stage detector, across varying compression ratios from 0.25% to 5.0%. The results demonstrate that performance on RetinaNet is comparable to that on Faster R-CNN across all IPDs. Specifically, at 0.25% IPD, $mAP_{50}$ for the Faster R-CNN observer and RetinaNet target configuration reaches 25.70%, surpassing the 24.70% obtained in the Faster R-CNN observer and target setup. At higher compression ratios, such as 1.0% and 5.0%, RetinaNet continues to yield competitive results, achieving $mAP_{50}$ scores of 37.40% and 48.60%, respectively. These findings further highlight the robustness of the distilled datasets, showcasing their effective applicability across different object detection architectures.

**Method Components.** Table 5 presents an ablation study evaluating the impact of candidate selection and candidate screening on the MS COCO performance across varying compression ratios (IPD). The table entries where both are not used correspond to when all objects from the training set are randomly assigned a location and inserted into the distilled images without any filtration. The results demonstrate the effectiveness of both components in improving the quality of the synthesized dataset. The addition of candidate screening further improves the results across all compression ratios. For example, there is 3.5% and a 5.7% increase in $mAP$ and $mAP_{50}$ for the 1.0% distilled dataset.

## 5. Conclusion

In this work, we introduced a new OD³ framework for optimization-free dataset distillation in object detection, achieving significant improvements over existing methods. Using a novel two-stage process of *candidate selection* and *candidate screening* driven by a pre-trained observer model, our framework strategically synthesized compact yet highly effective datasets tailored for object detection. Our method consistently demonstrated superior performance across multiple evaluation metrics. For instance, OD³ achieved more than 14.0% improvement in mAP₅₀ compared to the state-of-the-art method DCOD on MS COCO, further highlighting the efficacy of our optimization-free approach.

## Impact Statement

This paper presents work whose goal is to advance the field of Machine Learning by introducing $\mathrm{OD}^3$, an optimization-free approach to dataset distillation for object detection. Our method enhances efficiency and scalability, making high-quality dataset distillation more accessible while reducing computational costs. By simplifying the dataset generation process, we enable the development of more efficient models that can be deployed in resource-constrained environments. There are many potential societal consequences of our work, such as improving the accessibility of powerful machine learning models for industries with limited computational resources, and fostering the development of sustainable AI.

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

# Appendix

## A. Distilled Data Distribution Statistics

Table 6 presents the distribution of images and objects across 12 supercategories in MS COCO under different `IPD` settings as well as in the original dataset. The full dataset (100%) contains 64,115 images and 262,465 person-related objects, while the most compressed version (0.25%) retains only 295 images and 5,313 objects of this supercategory. Similar reductions are observed across all supercategories, demonstrating the significant compression effect of the **OD³** distillation process. We also present the ratio of the number of images in a particular supercategory at a certain compression ratio compared to the number of images of the corresponding supercategory in the original MS COCO dataset in Fig. 7.

Fig. 8 further illustrates the relative probability distribution of supercategories across dataset versions. Despite significant compression that can be seen in Table 6, the distribution remains statistically consistent with the original dataset. This shows that **OD³** does not introduce any inherent bias toward any specific category.

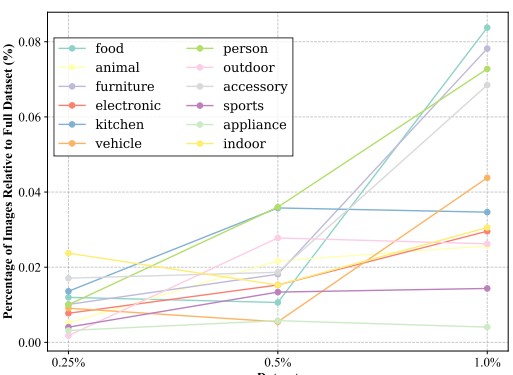

*Figure 7.* **Percentage of images in the distilled datasets relative to the full dataset.**

| IPD | Type | Supercategory in MS COCO | | | | | | | | | | | |
|---|---|---|---|---|---|---|---|---|---|---|---|---|---|
| | | **person** | **indoor** | **food** | **kitchen** | **appliance** | **furniture** | **vehicle** | **animal** | **electronic** | **accessory** | **outdoor** | **sports** |
| 100% | Images | 64115 | 15847 | 16255 | 20792 | 7880 | 29481 | 27358 | 23989 | 12944 | 17691 | 12880 | 23218 |
| (Full Dataset) | Objects | 262465 | 46088 | 63512 | 86677 | 13479 | 76985 | 96212 | 62566 | 28029 | 45193 | 27855 | 50940 |
| 1.0% | Images | 1183 | 733 | 660 | 948 | 260 | 1012 | 694 | 596 | 615 | 882 | 464 | 423 |
| | Objects | 20158 | 2876 | 5956 | 5922 | 831 | 5489 | 6035 | 4138 | 2047 | 3083 | 1407 | 1308 |
| 0.5% | Images | 585 | 366 | 313 | 463 | 120 | 496 | 351 | 279 | 270 | 433 | 219 | 212 |
| | Objects | 10257 | 1570 | 2963 | 2996 | 335 | 2486 | 3032 | 2142 | 848 | 1374 | 752 | 595 |
| 0.25% | Images | 295 | 187 | 155 | 221 | 50 | 242 | 189 | 142 | 137 | 220 | 115 | 94 |
| | Objects | 5313 | 712 | 1393 | 1636 | 113 | 1228 | 1513 | 985 | 479 | 773 | 360 | 245 |

*Table 6.* **Supercategory distribution across different `IPD` settings.** The number of images and objects per supercategory is presented for the MS COCO (Lin et al., 2014) dataset and the **OD³** distilled versions. Note that the data in the table is the same as in Fig. 7, but Fig. 7 displays the values as percentages. It can be seen that both the number of images and objects per supercategory are drastically compressed. Supercategory-level data is provided instead of fine-grained categories to maintain clarity and simplify comparisons.

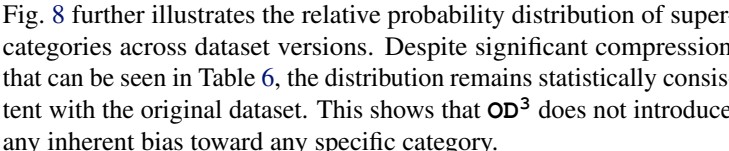
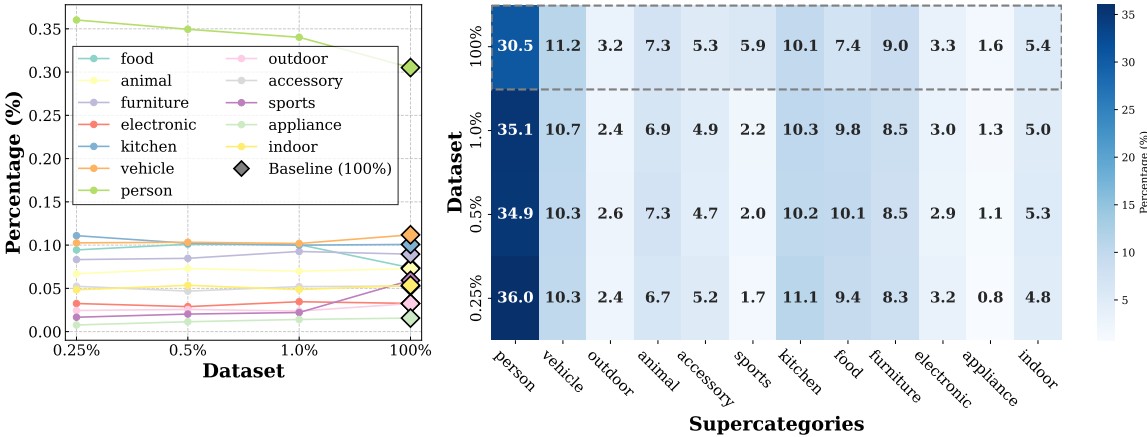

*Figure 8.* **Probability distribution of supercategories across datasets.** The figure highlights the relative distribution of each supercategory in the original MS COCO dataset (100%) and its synthesized counterparts at different compression ratios (0.25%, 0.5%, and 1.0%). This analysis provides shows that the synthesis process successfully mirrors the distribution of supercategories, $\mathcal{N}(x)$, in the original dataset.

# B. More Ablations

Table 7 further illustrates the impact of SA-DCE on object detection performance. Our SA-DCE method consistently outperforms both our no-extension and static extension baseline methods. Notably, it improves mAP scores while striking a balance between small and medium object detection. The no-extension setting suffers from reduced performance on small objects due to limited contextual information, whereas static extension provides slight improvements but lacks adaptability to object scale. In contrast, SA-DCE dynamically adjusts the context extension based on object size, leading to significant gains, particularly in small-object detection. These results demonstrate that SA-DCE effectively enhances detection robustness while preserving overall performance across different object scales.

| IPD | Extension (pixels) | mAP | $mAP_{50}$ | $mAP_{75}$ | $mAP_s$ | $mAP_m$ | $mAP_l$ |
|---|---|---|---|---|---|---|---|
| **0.5%** | No extension | 16.60 | 30.30 | 16.30 | 6.80 | 21.70 | **23.30** |
| | Static extension | 16.80 | 30.70 | 16.60 | 7.40 | 22.10 | 22.90 |
| | SA-DCE | **17.40** | **32.10** | **17.00** | **8.40** | **23.00** | 22.80 |
| **1.0%** | No extension | 22.00 | 38.70 | 22.30 | 9.70 | 27.40 | 30.10 |
| | Static extension | 22.30 | 38.80 | 22.90 | 9.90 | 27.40 | **30.40** |
| | SA-DCE | **22.50** | **39.60** | **22.90** | **10.60** | **28.00** | 29.80 |

*Table 7.* **Ablation Study of SA-DCE.** The table evaluates the influence of extending statically and dynamically (using SA-DCE) the bounding boxes (in pixels) of the objects in the distilled dataset across varying compression ratios on the MS COCO performance. Static extension refers to applying constant extension $\bar{r}$ to all inserted objects regardless of their size. We set $\bar{r}$ as 10 pixels.

Table 8 highlights the effect of varying the confidence threshold ($\eta$) on detection performance. Setting $\eta = 0.2$ consistently yields the best overall results across different IPD values, improving mAP and balancing small, medium, and large object detection. Lower thresholds ($\eta = 0.1$) allow more candidates but introduce noise, while higher thresholds ($\eta \geq 0.3$) remove potentially useful detections, leading to a drop in performance. These findings demonstrate that careful tuning of $\eta$ is crucial for optimizing detection accuracy.

| IPD | Confidence Threshold ($\eta$) | mAP | $mAP_{50}$ | $mAP_{75}$ | $mAP_s$ | $mAP_m$ | $mAP_l$ |
|---|---|---|---|---|---|---|---|
| **0.25%** | 0.1 | 11.10 | 21.70 | 10.50 | 5.30 | 15.50 | 14.50 |
| | 0.2 | **12.80** | **24.70** | **11.90** | **5.60** | **16.80** | **17.70** |
| | 0.3 | 10.50 | 21.00 | 9.40 | 4.90 | 14.80 | 13.40 |
| | 0.4 | 10.00 | 19.80 | 9.00 | 5.40 | 13.80 | 12.90 |
| | 0.5 | 10.20 | 20.30 | 9.30 | 5.00 | 14.50 | 12.90 |
| **0.5%** | 0.1 | 17.00 | 31.50 | 16.60 | 8.10 | 22.80 | 22.60 |
| | 0.2 | **17.40** | **32.10** | **17.00** | **8.40** | **32.00** | **22.80** |
| | 0.3 | 16.20 | 30.10 | 15.90 | 7.30 | 21.60 | 21.60 |
| | 0.4 | 16.40 | 30.60 | 16.10 | **8.40** | 22.50 | 21.90 |
| | 0.5 | 15.70 | 29.50 | 15.20 | 7.20 | 21.30 | 20.50 |
| **1.0%** | 0.1 | 21.70 | 38.30 | 22.20 | **10.80** | 27.80 | 28.90 |
| | 0.2 | **22.50** | **39.60** | **22.90** | 10.60 | **28.00** | **29.80** |
| | 0.3 | 22.00 | 39.00 | 22.30 | 10.60 | 27.60 | 29.30 |
| | 0.4 | 22.00 | 38.80 | 22.40 | 10.20 | 27.80 | 28.90 |
| | 0.5 | 21.80 | 38.50 | 22.20 | 10.30 | 27.70 | 29.00 |

*Table 8.* **Ablation Study of Confidence Threshold ($\eta$).** Objects with confidence lower than $\eta$ (determined by observer model) are removed in the candidate screening stage.

Table 9 analyzes the effect of canvas size on detection performance across different IPD values. The canvas size was selected based on the average width and height of all training images in the MS COCO dataset, with additional smaller and larger canvas sizes included for comparison and evaluation. Results indicate that while an optimal canvas size ($484 \times 578$) achieves

the highest mAP scores, further reduction in canvas dimensions leads to a drop in performance. This suggests that excessively small canvases limit the available contextual information, negatively impacting detection accuracy. Conversely, overly large canvases introduce unnecessary background noise, reducing effectiveness. These findings highlight the importance of selecting a balanced canvas size to maximize object representation while maintaining relevant contextual cues for dataset distillation.

| IPD | Canvas Size (pixels) | mAP | mAP$_{50}$ | mAP$_{75}$ | mAP$_s$ | mAP$_m$ | mAP$_l$ |
|---|---|---|---|---|---|---|---|
| **0.25%** | $363 \times 433$ | 10.30 | 20.80 | 9.10 | 4.80 | 13.60 | 14.20 |
| | $484 \times 578$ | **12.80** | **24.70** | **11.90** | **5.60** | **16.80** | **17.70** |
| | $726 \times 867$ | 10.50 | 20.50 | 9.50 | 4.70 | 16.30 | 12.50 |
| | $968 \times 1156$ | 8.80 | 17.50 | 7.80 | 3.60 | 14.90 | 10.20 |
| **0.5%** | $363 \times 433$ | 15.40 | 29.10 | 14.70 | 7.20 | 19.90 | 21.50 |
| | $484 \times 578$ | **17.40** | **32.10** | **17.00** | **8.40** | **23.00** | **22.80** |
| | $726 \times 867$ | 15.70 | 29.10 | 15.20 | 7.30 | 22.60 | 19.90 |
| | $968 \times 1156$ | 13.70 | 26.00 | 12.90 | 7.00 | 20.00 | 16.60 |
| **1.0%** | $363 \times 433$ | 20.90 | 37.20 | 21.10 | 9.80 | 25.70 | 28.30 |
| | $484 \times 578$ | **22.50** | **39.60** | **22.90** | 10.60 | **28.00** | **29.80** |
| | $726 \times 867$ | 21.00 | 37.30 | 21.40 | **10.80** | 27.60 | 26.40 |
| | $968 \times 1156$ | 16.80 | 30.40 | 16.90 | 8.40 | 24.10 | 20.40 |

*Table 9.* **Ablation Study of Canvas Size.** The table evaluates the influence of canvas size on the MS COCO performance of the distilled dataset across varying compression ratios.

## C. Proof of Theorem 3.2

*Proof.* Let $t \in \mathbb{N}$ be the current iteration index with $0 \leq t < T$. We assume for this iteration that objects placed on the canvas $\mathbf{x}_t$ do not overlap. Let the canvas $\mathbf{x}_t$ contain $K$ objects $\{o_r\}_{r=0}^K$. We sort these objects according to their confidence scores $q(o_r)$ and partition them into two sets based on a threshold $\eta$:

$$\begin{aligned} \{o_{i_0}, o_{i_1}, \ldots, o_{i_M}\} & \quad \text{where} \quad q(o_{i_0}) \leq q(o_{i_1}) \leq \cdots \leq q(o_{i_M}) < \eta, \\ \{o_{i_{M+1}}, o_{i_{M+2}}, \ldots, o_{i_K}\} & \quad \text{where} \quad \eta \leq q(o_{i_{M+1}}) \leq \cdots \leq q(o_{i_K}). \end{aligned} \tag{17}$$

The first set satisfy $\mathbb{E}_r[p(q(o_{i_r}) < \eta)] = \frac{M+1}{K}$. Applying the removal operator $f_{\text{remove}}(\mathbf{x}_t)$ discards every object whose confidence is below $\eta$, i.e., $\{o_{i_0}, o_{i_1}, \ldots, o_{i_M}\}$.

Hence, the updated canvas $\mathbf{x}_{t+1}$ preserves only those objects whose scores exceed $\eta$, and it may then be "refilled" by $f\text{add}(\cdot)$ with new (randomly synthesized) objects from the same distribution as in previous iterations.

Then, we can compare the $\Phi(\mathbf{x})$ of $\mathbf{x}_t$ and $\mathbf{x}_{t+1}$. $\Phi(\mathbf{x})$ can be described as

$$\Phi(\mathbf{x}) = \frac{\sum_{j=0}^M s(o_{i_j}) q(o_{i_j}) + \sum_{j=M+1}^K s(o_{i_j}) q(o_{i_j})}{\sum_{j=0}^M s(o_{ij}) + \sum_{j=M+1}^K s(o_{i_j})} \tag{18}$$

where $\sum_{j=M+1}^K s(o_{i_j})$ and $\sum_{j=M+1}^K s(o_{i_j}) q(o_{i_j})$ are same for $\mathbf{x}_t$ and $\mathbf{x}_{t+1}$. In general, we will fill the canvas at each iteration, so $\sum_{j=0}^M s(o_{i_j})$ can also be considered constant. And the difference between $\mathbf{x}_t$ and $\mathbf{x}_{i+1}$ is $\frac{\sum_{j=0}^M s(o_{i_j}) q(o_{i_j})}{S}$, where $S$ is the areas of the canvas. Due to $\mathbb{E}_{0 \leq j \leq M}[p(q(o_{i_j}) < \eta)] = 1$ for $\mathbf{x}_t$, we can get $p(\mathbb{E}_{0 \leq j \leq M}[q(o_{i_j})] - \mathbb{E}[\eta] \geq 0) = 0$, and $\mathbb{E}_{0 \leq j \leq M}[p(q(o_{i_j}) < \eta)] = \frac{M+1}{K}$ for $\mathbf{x}_{t+1}$. Then, we can get

$$p(\mathbb{E}_{0 \leq j \leq M}[q(o_{i_j})] - \mathbb{E}[\eta] \geq 0) = \frac{K - M - 1}{K} \tag{19}$$

Since object is uniformly distributed, so $p$ and $\mathbb{E}$ are able to swap places. Because $p\left(\mathbb{E}_{0\leq j\leq M}\left[q\left(o_{i_j}\right)\right] \geq \eta\right) = \frac{K-M-1}{K}$ for $\mathbf{x}_{i+1}$, we can prove that

$$\Phi\left(\mathbf{x}_T\right) \geq \Phi\left(\mathbf{x}_{T-1}\right) \geq \Phi\left(\mathbf{x}_{T-2}\right) \geq \cdots \geq \Phi\left(\mathbf{x}_0\right) \tag{20}$$

**Consistency of $\mathcal{N}(\mathbf{x})$ and overlaps.** As $T \to \infty$, the canvas becomes fully populated in both the *add-only* and *add-then-remove* strategies, so the number of objects $\mathcal{N}\left(\mathbf{x}_T\right)$ is generally similar (i.e., both can fill the canvas to full capacity).

*Overlap handling.* When $T$ is sufficiently large, the canvas will necessarily be filled, so it can be assumed that the first form and the second form of $\mathcal{N}\left(\mathbf{x}_T\right)$ are consistent. So $G_2$ remains greater than $G_1$. When there are some overlaps of objects in the iteration, the conclusion still holds. For example, object $o_a$ and $o_b$ overlap, and their overlap region is $o_d$. The score of $o_d$ is between $q\left(o_a\right)$ and $q\left(o_b\right)$. If both $q\left(o_a\right)$ and $q\left(o_b\right)$ are larger or smaller than $\eta$, then both of them will not be considered. If one is larger and one smaller than $\eta$ (assuming that $q\left(o_a\right) \leq \eta$ and $q\left(o_b\right) \geq \eta$), then $o_a$ is removed and $o_c$ will also be removed in the process of *screening*, and the portion left behind (i.e., possibly the mutilated $o_b \to \hat{o}_b$) may not be detectable by the detector, or it may be successfully detected. Even assuming that this is undetectable for $\hat{o}_b$ (i.e., the confidence score is low), then in the next iteration it will still be removed.

Assume that the probability of having no overlap with another object is $p_1$. The probability that $q\left(\hat{o}_b\right) \leq \tau$ is detected will be $p_2$. This probability of it being removed or not having an overlap in the next iteration is $p_1 + \left(1 - p_1\right) p_2$, which is consistently greater than $\left(1 - p_1\right)\left(1 - p_2\right)$ when $p_2 \geq 0.5$. If $p_2 < 0.5$, this means that $\hat{o}_b$ is a qualified sample (detectable by detector or observer) and therefore does not need to be removed.

Thus, in the presence of overlap, $G_2$ remains greater than $G_1$.

$\square$

