# OpenReview forum: "OD³: Optimization-free Dataset Distillation for Object Detection"
_ICML.cc/2025/Conference — Submitted to ICML 2025_

### Official Review · Reviewer_LWSm · 2025-03-10

**Overall Recommendation:** 2

**Summary:**

This work presents an optimization-free data distillation framework for object detection. It addresses the challenges of training large neural networks on large-scale datasets by synthesizing compact datasets. The framework consists of two main stages: candidate selection, where object instances are iteratively placed in synthesized images, and candidate screening, where a pre-trained observer model removes low-confidence objects. Experiments on MS COCO and PASCAL VOC datasets with compression ratios ranging from 0.25% to 5% show that the proposed method outperforms existing methods, achieving significant improvements in accuracy.

## update after rebuttal
While many studies report AP50 for the VOC dataset, mAP remains a valid and comprehensive metric for evaluating object detection models. The authors did not address my concern regarding why mAP is higher than mAP50 on VOC. The authors responded that mAP is redundant and AP50 is the standard practice for VOC, but this does not directly resolve my question.

Additionally, soft labels at the feature level would require significant storage. Traditional soft labels for images already demand substantial storage—often exceeding the storage of the entire dataset itself. Thus, applying soft labels at the feature level further exacerbates the unfairness in dataset distillation.

**Claims And Evidence:**

Yes

**Essential References Not Discussed:**

No.

**Experimental Designs Or Analyses:**

1. The comparison on the benchmarks is unfair for the baseline method including random, uniform, k-center, herding and DCOD. All the results are directly copied from the DCOD (NeurIPS 2024). These methods use the YOLO V3 as the base detector while the proposed method uses the Faster-RCNN 50, maybe even with FPN. Therefore, the experiment comparison in Table 1 and Figure 1 are unfair.
2. In Table 2, for the PASCAL VOC dataset, why do the authors only present the results of the proposed method OD3 and not include the baselines?

**Methods And Evaluation Criteria:**

Yes

**Other Comments Or Suggestions:**

1. The algorithm needs to be more clear. how can a bounding box $b_{ir}$ add $l_{ir}? $\hat{x}$ is nor defined.
2. In Eq.(2), why can the proposed method represent the information density?

**Other Strengths And Weaknesses:**

Strengths:

1. The proposed method is optimize-free. By avoiding computationally intensive iterative optimization processes, the proposed method is more efficient. The data generation process is highly efficient as it does not require training, and the main time overhead comes from the screening by the observer model.
2. The distilled datasets can generalize well across different object detection architectures.


Weaknesses:
1. The experiment comparison is unfair as referred to in the comments about "Experimental Designs Or Analyses".
2. In Table 2, the results are very surprising. The proposed method achieves impressive mAP on the Pascal VOC dataset. When IPD=0.5%, the mAP is larger than mAP50, which is the opposite of the traditional observation. Usually, the metric mAP is much smaller than mAP50.
3. Lack of ablation study about the soft-label generation.
4. How about the storage of the soft labels? In many dataset distillation for classification works, researchers found that the soft-label will take too much storage. If the structure of the detector is different, how do we use the feature-based soft labels?

**Questions For Authors:**

1. The function 'a' in Eq.(2) and (3) are conflicts.
2. In Eq.(4), the Information Diversity is the number of distinct objects on the canvas. How do the authors define the distinct objects?

**Relation To Broader Scientific Literature:**

This work is highly related to the area of data efficiency and extends the dataset distillation into object detection beyond classification.

**Theoretical Claims:**

The theoretical of theorems 2 show the proposed method maintains the information density and diversity.

---

> ### Author Rebuttal · Authors · 2025-03-31
>
> Thank you for your valuable suggestions and for giving us the opportunity to address the points you raised!
>
> >**Q1: The experiment comparison is unfair for the baseline methods (random, uniform, k-center, herding, and DCOD).**
>
> We appreciate this concern. Below are our own runs of the core-set selection methods using our framework with 1% IPD:
>
>
> | Method   | mAP | mAP@50 | mAP@75 | mAP$_{s}$| mAP$_{m}$ | mAP$_{l}$|
> |----------|--------|--------|--------|--------|--------|--------|
> | **Random**   | 8.8    | 19.6   | 6.8    | 3.6    | 10.0   | 11.3   |
> | **Herding**  | 8.3    | 18.0   | 6.5    | 4.4    | 10.1   | 9.8    |
> | **K-center** | 9.20   | 20.20  | 7.10   | 4.30   | 10.20  | 12.30  |
> | **Uniform**  | 8.5    | 19.4   | 6.1    | 3.1    | 9.7    | 10.8   |
>
> **DCOD [1]**: Since their codebase is not publicly available, we contacted the authors to share the codebase or the datasets to facilitate better comparison with their method and exact reproduction of their results. However, they have declined to share them. Also, YOLOv3-SPP (their baseline) outperforms Faster R-CNN 101 (our observer baseline) with reported performance of 44.3% vs. 39.8%, respectively. We believe that the comparison remains fair, as our approach is evaluated under a more challenging setting (lower-performing baseline).
>
> >**Q2: For the PASCAL VOC dataset, why do the authors only present the results of the proposed method and not include the baselines?**
>
> For the PASCAL VOC dataset, we are unable to directly compare against DCOD since they used COCO metrics instead of the standard PASCAL metrics, which does not facilitate a direct comparison with training on the uncompressed dataset. However, we will include the results for the baseline core-set selection methods as follows (1.0% IPD):
>
> | Method   | mAP@50 |
> |-------|------|
> | **Random**   | 38.20  |
> | **Herding**  | 35.00  |
> | **K-center** | 36.60  |
> | **Uniform**  | 37.25  |
>
> >**Q3: In Table 2, the results are very surprising. Usually, the mAP metric is much smaller than mAP50.**
>
> We appreciate the reviewer's keen observation. Upon further inspection, the reported metrics are redundant, and we will keep the mAP@50 column, which reflects the default VOC metric using the area method.
>
> >**Q4: Lack of ablation study on soft-label generation.**
>
> As suggested, we will add the below results of the soft-label ablation:
> | Method         | mAP  | mAP@50 | mAP@75 | mAP$_{s}$| mAP$_{m}$ | mAP$_{l}$ |
> |-----------|------|--------|--------|------|------|------|
> | **Without soft label** | 17.1 | 31.4   | 17.0   | 6.0  | 19.0 | 24.1 |
> | **With soft label**  | 22.5 | 39.6   | 22.9   | 10.6 | 28.0 | 29.8 |
>
> >**Q5: How is the storage of soft labels handled? In many dataset distillation works for classification, researchers found that soft labels require significant storage.**
>
> Following the approach of RDED [2], we avoid explicitly storing soft labels by using a knowledge distillation framework with a teacher model.
>
> >**Q6: The algorithm needs clarification. How can a bounding box b add ℓ? Also, $\hat{x}$ is not defined in the algorithm.**
>
> We understand the confusion and will re-define the bounding box modification as follows:
>
> $b' = (x - \ell_x, y - \ell_y, w + 2\ell_x, h + 2\ell_y)$
>
> As for $\hat{x}$, we will define it as the final reconstructed image after adding the objects to the canvas.
>
>
> >**Q7: In Eq. (2), how does the proposed method represent information density?**
>
> The object confidence score normalized by area effectively captures the information density of the canvas. Based on this formulation, we select objects with higher confidence scores. A higher value indicates that the canvas contains more object-related information, while the object size ensures that confidence is measured per unit area.
>
>
> >**Q8: The function 'a' in Eq. (2) and Eq. (3) conflicts.**
>
> Originally, we used a different symbol in Eq. (3), so to resolve the conflict, we will change 'a' in Eq. (2) to uppercase or adopt an alternative notation for clarity.
>
>
> >**Q9: In Eq. (4), the Information Diversity metric counts distinct objects on the canvas. How are distinct objects defined?**
>
> As objects are added to the canvas, the method tracks each object {or} separately, even if it intersects with other objects. This ensures that distinct objects are accounted for.
>
>
> We appreciate your thoughtful feedback and hope these clarifications address your concerns.
>
>
>
> **References:**
>
> [1] Qi, Ding, Jian Li, Jinlong Peng, Bo Zhao, Shuguang Dou, Jialin Li, Jiangning Zhang, Yabiao Wang, Chengjie Wang, and Cairong Zhao. "Fetch and forge: Efficient dataset condensation for object detection." Advances in Neural Information Processing Systems 37 (2024): 119283-119300.
>
> [2] Sun, Peng, Bei Shi, Daiwei Yu, and Tao Lin. "On the diversity and realism of distilled dataset: An efficient dataset distillation paradigm." In Proceedings of the IEEE/CVF Conference on Computer Vision and Pattern Recognition, pp. 9390-9399. 2024.

---

> > ### Comment · Reviewer_LWSm · 2025-04-02
> >
> > Thanks for the detailed response. The response only addressed part of my concerns.
> >
> > 1. For Q3, what is the meaning of "the reported metrics are redundant"? MAP is also an important metric for evaluating detection performance beyond mAP50.
> > 2. The soft label claimed in this paper is actually the features, not the labels.

---

> > > ### Author Response · Authors · 2025-04-08
> > >
> > > We thank you for your response! We would like to clarify a few points regarding the concerns raised.
> > >
> > > **Q1: For Q3, what is the meaning of "the reported metrics are redundant"? MAP is also an important metric for evaluating detection performance beyond mAP50.**
> > >
> > > The official Pascal VOC evaluation protocol strictly reports box AP at 50% IoU (mAP@50) rather than the COCO-style mAP averaged over IoUs from 0.5 to 0.95. The VOC dataset is relatively simpler than COCO, as it typically contains fewer objects per image and those objects tend to be larger. In this context, evaluating detection performance using AP at 50% IoU (mAP@50) is more appropriate and meaningful. Consequently, most prior studies on VOC do not report the COCO-style mAP (averaged over IoUs from 0.5 to 0.95), as it offers limited practical value for this dataset.
> > >
> > > **Q2: The soft label claimed in this paper is actually the features, not the labels.**
> > >
> > > We have experimented with both traditional soft labels and feature-based/channel-wise soft labels and found that the current approach yields the most significant performance improvement for object detection tasks. We have discussed and elaborated on this in detail in Section 3.2 from lines 219 to 266.
> > >
> > > We hope this clarification addresses the concerns raised, and we appreciate the constructive feedback that helps improve our work.

---

### Official Review · Reviewer_tJdV · 2025-03-13

**Overall Recommendation:** 4

**Summary:**

The work proposes a new framework called OD3 (Optimization-free Dataset Distillation for Object Detection), specifically designed for dataset distillation in object detection tasks. It aims to reduce training time and computational resources by selecting and generating a high-quality compact dataset from a large dataset to replace the large dataset. The distillation process consists of two stages: 1. Candidate Selection, where potential candidates with high information density and diversity are chosen, representing the most valuable information in the original dataset; 2. Candidate Screening, where further screening of the selected candidates is conducted to ensure that the final generated dataset can effectively represent the spatial and semantic characteristics of the original dataset. Information density and information diversity are used as selection criteria to ensure that the selected samples can maximally represent the original dataset. SA-DCE is proposed to address the scale variation and spatial layout issues specific to object detection. Experiments were conducted on two widely used datasets, MS COCO and PASCAL VOC.

**Claims And Evidence:**

The OD3 framework distills a small-scale dataset from a large dataset while maintaining model performance. In the experimental section, experiments were conducted with different compression ratios, the SA-DCE module, and various overlap thresholds. The experiments demonstrate that detectors trained based on the OD3 algorithm outperform other algorithms, essentially proving the argument.
In addition, the right of line 102 mentions that this work does not sacrifice performance significantly, but the experiment results suggest the performance drop is somewhat significant.

**Essential References Not Discussed:**

All key references have been cited.

**Experimental Designs Or Analyses:**

1. How does this method perform on the latest transformer-based detectors?
2. Additional ablation experiments regarding the confidence threshold eta are missing.
3. The dataset distillation technique claims to speed up training while maintaining model performance. Could the authors provide a performance comparison with models trained on uncompressed datasets?

**Methods And Evaluation Criteria:**

OD3 is a novel optimization-free dataset distillation framework designed specifically for object detection. It involves two main stages: candidate selection and candidate screening. In the first stage, object instances are randomly placed on a blank canvas, ensuring minimal overlap. In the second stage, a pre-trained observer model evaluates and filters out low-confidence objects. The framework also uses scale-aware dynamic context extension to enhance small object detection by expanding the bounding areas based on object size. This approach allows OD3 to generate compact, high-fidelity datasets efficiently, significantly reducing the dataset size while maintaining or even improving detection performance compared to existing methods.

**Other Comments Or Suggestions:**

No other comments.

**Other Strengths And Weaknesses:**

The proposed method significantly improves training efficiency but requires further experiments to confirm that the model's performance is maintained before and after distillation.
In data distillation, Faster RCNN101 was used as the observer model. However, the accuracy of the model trained based on distillation data collection was only 30.1, which could not even reach the accuracy of the observer model and could not clearly reflect the significance of distillation data.
Moreover, the performance of Faster RCNN101 and Faster RCNN50 on COCO is not reported in the paper.

**Questions For Authors:**

No other questions.

**Relation To Broader Scientific Literature:**

The main contribution of the OD3 framework is closely related to the broader scientific literature, addressing the specific needs of dataset distillation for object detection, a task often overshadowed by image classification in previous work. Traditional methods heavily rely on optimization, while OD3 introduces an optimization-free approach, leveraging the concepts of information density and diversity from active learning to ensure a representative and compact dataset. Additionally, the introduction of Scale-Aware Dynamic Context Expansion (SA-DCE) addresses the challenges of scale variation in object detection, which were previously tackled by multi-scale techniques.

**Theoretical Claims:**

This work contains little proof or theoretical claims.

---

> ### Author Rebuttal · Authors · 2025-03-31
>
> We sincerely appreciate your thorough review and for recognizing our contributions!
>
> >**Q1: How does this method perform on the latest transformer-based detectors?**
>
> We agree that evaluating performance on transformer-based detectors is important. To further demonstrate the generalizability of our approach, we have conducted additional experiments and included the results accordingly.
>
> | IPD  | Observer Model    | Target Model  | mAP  | mAP@50 | mAP@75 |
> |------|------------------|--------------|------|--------|--------|
> | 0.25% | Deformable DETR | Faster RCNN  | 11.90 | 22.60  | 11.10  |
> | 0.5%  | Deformable DETR | Faster RCNN  | 16.20 | 29.50  | 16.00  |
> | 1.0%  | Deformable DETR | Faster RCNN  | 22.00 | 38.00  | 22.90  |
> | 0.5%  | DETR           | Faster RCNN  | 12.10 | 26.40  | 9.40   |
> | 1.0%  | DETR           | Faster RCNN  | 16.40 | 33.90  | 13.90  |
>
> >**Q2: Additional ablation experiments regarding the confidence threshold are missing.**
>
> Thank you for your suggestion. The ablation study on the confidence threshold is indeed provided in Table 8 of the appendix.
>
> >**Q3: The dataset distillation technique claims to speed up training while maintaining model performance. Could the authors provide a performance comparison with models trained on uncompressed datasets?**
>
> Certainly! The following is the reported performance by mmdetection of the used models on the full versions of MS COCO:
>
> | Model        | mAP  | mAP@50 | mAP@75 |
> |--------------------|------|--------|--------|
> | Faster R-CNN 50    | 38.4 | 59.0   | 42.0   |
> | Faster R-CNN 101 | 39.8 | 60.1   | 43.3   |
> | RetinaNet 50       | 37.4 | 56.7   | 39.6   |
> | RetinaNet 101| 38.9 | 58.0   | 41.5   |
>
> >**Q4: The performance drop is somewhat significant between distilled training and full-scale training.**
>
> The performance drop is 8.9%, from 39% (full-scale with 100% of the original dataset) to 30.1% (distilled with only 5% IPD). With these results, we have successfully bridged around 10% of the gap between the previous SOTA and the theoretical upper bound at a 1.0% compression rate. Similarly, a recent work from the image classification task, which has been more thoroughly explored, is RDED [1]. This method achieved 33.9% accuracy at IPC=10 on ImageNet-100 with ResNet-101 vs training full-scale on the entire dataset achieving 78.25% (gap of 44.35%). Thus, we believe $OD^3$ exhibits a huge step forward toward completely bridging the gap between distilled and full-scale training.
>
> We appreciate your valuable feedback and hope this addresses your concerns.
>
> **References:**
>
> [1] Sun, Peng, Bei Shi, Daiwei Yu, and Tao Lin. "On the diversity and realism of distilled dataset: An efficient dataset distillation paradigm." In Proceedings of the IEEE/CVF Conference on Computer Vision and Pattern Recognition, pp. 9390-9399. 2024.

---

### Official Review · Reviewer_ZDm5 · 2025-03-14

**Overall Recommendation:** 3

**Summary:**

The paper proposes a dataset distillation method for object detection datasets, aiming at condensing the number of training images down to 0.25 - 5% of the original training dataset. This is achieved by first copy-pasting objects from the training set onto blank backgrounds. In a second step, objects that are assigned low confidence by a pre-trained model are removed. Finally, a target model is distilled from soft labels output by the larger pre-trained model. The effectiveness of the method is evaluated on COCO and Pascal VOC and outperforms prior work on these benchmarks.

**Claims And Evidence:**

The paper's claims are supported by sufficient evidence. The evaluation demonstrates substantial improvements in effectiveness over prior methods and the ablation studies clearly demonstrate the contributions of label types, the candidate selection, and the candidate screening components.

**Essential References Not Discussed:**

None.

**Experimental Designs Or Analyses:**

The experimental design is in line with prior work and uses (distilled versions of) COCO and Pascal VOC as standard benchmark datasets for object detection. The metrics used (mAP at different IoU thresholds) make sense as well.

**Methods And Evaluation Criteria:**

While in essence a simple extension of copy-paste combined with knowledge distillation, the proposed method makes sense overall. To the best of my knowledge, the paper compares the method's effectiveness to the (few) relevant prior works, which are outperformed by a significant margin. However, the evaluation is limited to a single backbone architecture (ResNet-50) and two detectors (RetinaNet & Faster R-CNN), which severely limits its generality. Since the proposed method is in principle maximally general and the key promise is to reduce training effort, I think the paper should additionally evaluate the method on more modern backbones (such as ViTs) and detectors (such as DETR).

**Other Comments Or Suggestions:**

1. Eq. 6 has weird formatting
2. x_{i+1} is missing superscripts in Eq. 10 & Eq. 12
3. mmrazor reference renders as "(Contributors, 2021)"

**Other Strengths And Weaknesses:**

None.

**Questions For Authors:**

None.

**Relation To Broader Scientific Literature:**

The paper is concerned with the underexplored area of dataset distillation for object detection. The method is maximally simple, combining proven ideas such as copy-paste augmentation and knowledge distillation.

**Theoretical Claims:**

The main theoretical claim is that the proposed add-then-remove scheme of step-wise removing low-confidence pasted objects achieves a greater objective value than the add-only strategy. I have not checked this claim in detail but it makes intuitive sense.

---

> ### Author Rebuttal · Authors · 2025-03-31
>
> Thank you for your thoughtful review and valuable suggestions!
>
> >**Q1: The evaluation is limited to a single backbone architecture (ResNet-50) and two detectors (RetinaNet & Faster R-CNN), which severely limits its generality. The paper should additionally evaluate the method on more modern backbones (such as ViTs) and detectors (such as DETR).**
>
> We appreciate this insight and agree that evaluating performance on transformer-based models is important. To further demonstrate the generalizability of our approach, we have conducted additional experiments and included the results accordingly.
>
> | IPD   | Observer Model   | Target Model  | mAP   | mAP@50 | mAP@75 |
> |-------|-----------------|--------------|-------|--------|--------|
> | 0.25% | Deformable DETR | Faster RCNN  | 11.90 | 22.60  | 11.10  |
> | 0.5%  | Deformable DETR | Faster RCNN  | 16.20 | 29.50  | 16.00  |
> | 1.0%  | Deformable DETR | Faster RCNN  | 22.00 | 38.00  | 22.90  |
> | 0.5%  | DETR           | Faster RCNN  | 12.10 | 26.40  | 9.40   |
> | 1.0%  | DETR           | Faster RCNN  | 16.40 | 33.90  | 13.90  |
>
> >**Q2: Eq. 6 has formatting issues.**
>
> We will redefine the equation as follows:
>
> $\quad \mathbf{z}_i = f^\textrm{fpn}(f^\textrm{backbone}(\mathbf{x}_i))$
>
> With the updated format:
>
> $\mathcal{L}\textrm{mse} = \mathbb{E}{(\mathbf{x}_i,\mathbf{y}^\textrm{feat}_i)}
> \Big\Vert \mathbf{y}^\textrm{feat}_i - \frac{\mathbf{z}_i - \textrm{mean}(\mathbf{z}_i)}
> {\textrm{std}(\mathbf{z}_i) + \epsilon} \Big\Vert_2^2.$
>
>
>
>
>
> >**Q3: $x_{i+1}$  is missing superscripts in Eq. 10 & Eq. 12.**
>
> The superscripts $^{a}$ and $^{ar}$ will be added to $x_{i+1}$ in the revised version.
>
>
>
> >**Q4: The mmrazor reference renders as "(Contributors, 2021)".**
>
> As suggested, we will correct this reference in the revised version.
>
> We appreciate your valuable feedback and hope these clarifications address your concerns.

---

> > ### Comment · Reviewer_ZDm5 · 2025-04-03
> >
> > > We appreciate this insight and agree that evaluating performance on transformer-based models is important. To further demonstrate the generalizability of our approach, we have conducted additional experiments and included the results accordingly.
> >
> > I appreciate the additional experiments with DETR detectors. I assume these results are using ResNet backbones? I still think it would be a meaningful improvement to add experiments using ViT backbones instead.

---

> > > ### Author Response · Authors · 2025-04-08
> > >
> > > Thank you for the suggestion. We agree that evaluating ViT-based backbones provides meaningful insights. Accordingly, we have included additional experiments using ViTDet [1] with a ViT-B backbone, and we will report the results in the revised manuscript.
> > >
> > > | IPD     | Observer Model     | Target Model   | mAP  | mAP@50 | mAP@75 |
> > > |---------|--------------------|----------------|------|--------|--------|
> > > | 0.25%   | ViTDet (ViT-B)     | Faster RCNN    | 11.0 | 21.3   | 10.1   |
> > > | 0.5%    | ViTDet (ViT-B)     | Faster RCNN    | 15.9 | 29.2   | 15.6   |
> > > | 1.0%    | ViTDet (ViT-B)     | Faster RCNN    | 21.7 | 38.3   | 22.1   |
> > >
> > > **References:**
> > >
> > > [1] Li, Yanghao, et al. "Exploring plain vision transformer backbones for object detection." European conference on computer vision. Cham: Springer Nature Switzerland, 2022.

---

### Official Review · Reviewer_TmZx · 2025-03-14

**Overall Recommendation:** 3

**Summary:**

Dataset distillation for object detection is a under-explored task. This paper proposes a new optimization-free dataset distillation method tailored for object detection, named OD$^3$.  OD$^3$ consists of two steps: (1) an iterative candidate selection process that strategically places object instances in synthesized images; and (2) a candidate screening process powered by a pre-trained observer model, which discards low-confidence objects. Experiments on MS COCO and Pascal VOC datasets demonstrate the effectiveness of the proposed method.

**Claims And Evidence:**

The claims in the submission are supported by clear and convincing evidence.

**Essential References Not Discussed:**

N/A

**Experimental Designs Or Analyses:**

- In Table 1, the proposed method achieves significant performance improvements. However, the AP performance of different object sizes are omitted. It would be better to add more detailed comparison.
- In Table 5, the authors only evaluate RetinaNet and Faster R-CNN with ResNet backbone. All these models are somewhat outdated. How do the performance of detection transformers and more recent detectors perform?

**Methods And Evaluation Criteria:**

The proposed method is heuristic and somewhat trivial. It is hard to find some theoretical contributions or fresh insights.

**Other Comments Or Suggestions:**

N/A

**Other Strengths And Weaknesses:**

- This paper explores a under-explored task and achieves significant performance improvement.
- The overall paper is well organized and written.
- The proposed optimization-free approach shows promising efficiency.

**Questions For Authors:**

N/A

**Relation To Broader Scientific Literature:**

The idea of candidate screening is somewhat relevant to the context classifier in prior WSOD work [a]. The proposed object detection dataset distillation method may benefits to larger community.
[a] Object-aware instance labeling for weakly supervised object detection, ICCV'19

**Theoretical Claims:**

N/A

---

> ### Author Rebuttal · Authors · 2025-03-31
>
> Thank you for your constructive feedback and for giving us the opportunity to address your concerns!
>
> >**Q1: The AP performance for different object sizes is omitted.**
>
> The AP performance for different object sizes is reported in Table 3 and Table 5 of the main paper, as well as Table 7 and Table 8 of the appendix. Since these metrics were not reported for the SoTA method DCOD [1], we did not include them in Table 1 for direct comparison.
>
> >**Q2: How do detection transformers and more recent detectors perform?**
>
> We appreciate this suggestion and agree that evaluating transformer-based models is valuable. To further highlight the generalizability of our approach, we have conducted additional experiments and included the results accordingly.
>
> | IPD  | Observer Model    | Target Model  | mAP  | mAP@50 | mAP@75 |
> |------|------------------|--------------|------|--------|--------|
> | 0.25% | Deformable DETR | Faster RCNN  | 11.90 | 22.60  | 11.10  |
> | 0.5%  | Deformable DETR | Faster RCNN  | 16.20 | 29.50  | 16.00  |
> | 1.0%  | Deformable DETR | Faster RCNN  | 22.00 | 38.00  | 22.90  |
> | 0.5%  | DETR           | Faster RCNN  | 12.10 | 26.40  | 9.40   |
> | 1.0%  | DETR           | Faster RCNN  | 16.40 | 33.90  | 13.90  |
>
> We hope this clarifies your concerns, and we appreciate your thoughtful review!
>
> **References:**
>
> [1] Qi, Ding, Jian Li, Jinlong Peng, Bo Zhao, Shuguang Dou, Jialin Li, Jiangning Zhang, Yabiao Wang, Chengjie Wang, and Cairong Zhao. "Fetch and forge: Efficient dataset condensation for object detection." Advances in Neural Information Processing Systems 37 (2024): 119283-119300.

---

> > ### Comment · Reviewer_TmZx · 2025-04-05
> >
> > Thanks for your response. The response has addressed my concerns. I would like to raise my rating.

---

### Decision · Program_Chairs · 2025-05-01

**Decision:**

Reject

**Comment:**

This paper works on dataset distillation for object detection. It proposes an  optimization-free data distillation framework, including two stages: 1) a candidate selection process in which object instances are iteratively placed in synthesized images based on their optimal locations; and 2) a candidate screening process using a pre-trained observer model to remove low-confidence objects. The experiments have shown that the proposed method can achieve good results when the compression ratios ranges from 0.25% to 5%.

The main strengths include:
- This paper explores a under-explored task and achieves significant performance improvement.
- The proposed optimization-free approach shows promising efficiency.
- some claims are well supported with evidence.
- The overall paper is well organized and written.

The main concerns from the reviewers:
1. the proposed method is heuristic and somewhat trivial. It is hard to find some theoretical contributions or fresh insights.
2. the main idea is coy-paste combined with knowledge distillation from the literature.
3. the evaluated models are RetinaNet and Faster R-CNN with ResNet backbone, which are kind of outdated. Missing results on recent models.
4. significant performance drop when compared with fully trained models.
5. unfair comparison on the benchmark.
6. missing clarification on VOC eval and feature storage.

Although the rebuttal has resolved some concerns, some are not well resolved yet, including 1, 2, 6. Although the reviewers TmZx and ZDm5 are happy with the rebuttal on 3 and 4, the AC finds the responses are not very convincing. For example, for 3, the rebuttal only provides modern models as the teacher but not the target model, and for 4, the gaps to the fully trained models are indeed significant given the compression ratios. LWSm was not convinced by the rebuttal, although tJdV increased the score in the end.

Overall the AC is not convinced by the motivation of the paper. Although dataset distillation could be an interesting research direction, it is not clear 1) what's its benefits compared with weakly/semi-supervised learning, given they only need to use a small amount of supervision and they have better performance; and 2) what's the point to have low compression ratio if the performance drop is such significant. In addition, the technical contributions of this paper are incremental, given its similarity to copy-paste and knowledge distillation. So the AC thinks this paper is not good enough for ICML.